



# 1  Long-term trends in agricultural droughts over Netherlands and
# 2  Germany: how extreme was the year 2018?

Yafei Huang[1], Jonas Weis[2], Harry Vereecken[1], and Harrie-Jan Hendricks Franssen[1]
[1]Forschungszentrum Jülich, Agrosphere (IBG 3), Jülich, 52425, Germany
[2]The Hydrogeology Department, Karlsruhe Institute of Technology, 76131, Karlsruhe, Germany
*Correspondence to*: Yafei Huang (huangyafei136@yahoo.com)
**Abstract:** Droughts can have important impacts on environment and economy like in the year 2018 in parts of Europe.
Droughts can be analyzed in terms of meteorological drought, agricultural drought, hydrological drought and social-
economic drought. In this paper, we focus on meteorological and agricultural drought and analyzed drought trends for
the period 1965-2019 and assessed how extreme the drought year 2018 was in Germany and the Netherlands. The
analysis was made on the basis of the following drought indices: standardized precipitation index (SPI), standardized
soil moisture index (SSI), potential precipitation deficit (PPD) and ET deficit. SPI and SSI were computed at two time
scales, the period April-September and a 12-months period. In order to analyze drought trends and the ranking of the
year 2018, HYDRUS 1-D simulations were carried out for 31 sites with long-term meteorological observations and
soil moisture, potential evapotranspiration (ET) and actual ET were determined for five soil types (clay, silt, loam,
sandy loam and loamy sand). The results show that the year 2018 was severely dry, which was especially related to
the highest potential ET in the time series 1965-2019, for most of the sites. For around half of the 31 sites the year
2018 had the lowest SSI, and largest PPD and ET-deficit in the 1965-2019 time series, followed by 1976 and 2003.
The trend analysis reveals that meteorological drought (SPI) hardly shows significant trends over 1965-2019 over the
studied domain, but agricultural droughts (SSI) are increasing, at several sites significantly, and at even more sites
PPD and ET deficit show significant trends. The increasing droughts over Germany and Netherlands are mainly driven
by increasing potential ET and increasing vegetation water demand.

## 24  1 Introduction

Drought is a climatic phenomenon that is expected to increase in frequency and severity over Europe in
the future and is related to the ongoing climate change (Seneviratne et al., 2012a; Orlowsky and
Seneviratne, 2013; Mukherjee et al., 2018; Samaniego et al., 2018; Pokhrel et al., 2021). Droughts are
considered to be the most damaging natural hazard after floods, when measured globally in terms of the
population affected, suffering more than 2.2 million victims from 1950 to 2014 (Mishra and Singh, 2010;
Guha-Sapir, 2015; Zink et al., 2016). Droughts have a negative impact on food production, and shortage of
water supply, making it the costliest disaster in Europe in general (AghaKouchak, 2014; Guha-Sapir et al.,
2015; Zink et al., 2016).
We distinguish between meteorological, hydrological, agricultural and socio-economic drought, in
correspondence with Mishra and Singh (2010), Wilhite and Glantz (1985) and AMS (2004). Meteorological
drought is related to a period with rain deficiency with respect to the long-term average precipitation, and
is usually assessed by a standardized precipitation index (SPI) (AMS, 2004). Hydrological drought is
characterized by a river discharge deficit or low groundwater level (Wilhite and Glantz, 1985). Agricultural
drought is defined in terms of soil moisture deficit, leading to crop yield reductions (Anderson et al., 2016).
Agricultural drought and hydrological drought are affected by meteorological drought but not necessarily,
and with a certain delay with a typical time lag of a month or a few months. Hydrological drought and



agricultural drought usually lead to shortage of water supply for residents and jeopardize food security
and cause economic damages. Therefore, socio-economic drought is related to the mentioned three
drought indices.
Historical drought events like for the years 1976, 1996, 2003 and 2015 in Europe have been widely
investigated in literature, and it was concluded that these droughts had severe consequences for
agriculture, riverine transportation and drinking water (Sheffield and Wood, 2007; Mühr et al., 2018; Bakke
et al., 2020; Buitink et al., 2020; Buras et al., 2020). In Germany, huge economic losses have been reported
related to the record breaking temperatures and severe drought in the summer 2018 and this affected
agriculture, tourism and shipping losses (Mühr et al., 2018).
Analysis of droughts and studies on drought trends since the 1950s have been conducted (Sheffield and
Wood, 2007; Sheffield et al., 2012; Bakke et al., 2020). Barker studied droughts based on standardized
stream flow index for the period 1891 to 2015 for the UK and identified extreme droughts like the 1976
drought and also periods with multiple and longer droughts (e.g. early 1940s and early 1970s) at the
catchment scale (Barker et al., 2019). No consistent global drought trends were found and the trends
generally differ from region to region (Sheffield and Wood, 2007; Orlowsky and Seneviratne, 2013; Spinoni
et al., 2015). Sheffield and Wood (2007) showed that droughts increase in frequency and magnitude
globally but trends differ, depending on the drought index used. Orlowsky and Seneviratne (2013)
investigated the standardized precipitation index (SPI) as an indicator for meteorological drought for
observation datasets and simulations from the 5th phase of the Coupled Model Intercomparison Project
(CMIP5) and the results showed that frequency of droughts has increased in the Mediterranean, South
Africa and Central America/Mexico over the past decades and this trend would continue in the 21$^{st}$ century
according to CMIP projections (Orlowsky and Seneviratne, 2013). In Europe, the SPI-based analysis
indicates that droughts in northern Europe have been mitigated while in southern Europe the magnitude
of droughts have increased (Gudmundsson and Seneviratne, 2015). However, no consistent trends over
central Europe were found (Gudmundsson and Seneviratne, 2015; Spinoni et al., 2015).
In addition to drought trends, individual historic drought events also have been widely studied (Ciais et al.,
2005; Seneviratne et al., 2012b; Zink et al., 2016; Laaha et al., 2017; Bakke et al., 2020; Buitink et al., 2020;
Buras et al., 2020). For Central Europe the years 1976 and 2003 were affected by the most severe droughts
over the last 50 years (Ciais et al., 2005; Teuling et al., 2013). These two years gathered much attention
from media and were generally used as reference for recent drought events. The year 1976 suffered soil
moisture deficit and in central Europe the 2003 summer drought was more severe than the year 1976
(Teuling et al., 2013). The year 2003 witnessed a 30% reduction in primary productivity which was a
consequence of precipitation deficit and extreme heat waves (Ciais et al., 2005). Drought events in 1996
and 2015 were also analyzed in studies (Laaha et al., 2017; Barker et al., 2019). Barker showed that the
drought event 1995-1996 was the worst drought in central northern England since 1891 (Barker et al.,
2019). In 2015, drought also affected large parts of  Europe and especially the Eastern part of Europe
(Laaha et al., 2017). The year 2018 also drew much attention from media and scientists and is considered
to be one of the most extreme drought years and is analyzed and discussed in this paper.  From climatic-
data-based SPI it was found that northern Europe experienced an extreme drought in 2018 (Bakke et al.,
2020; Graf et al., 2020). Buitink et al. (2020) found, with help of an analysis based on in-situ data and
remote sensing data, that Netherlands witnessed in 2018 low soil moisture contents related to low
precipitation, which reduced ET (Buitink et al., 2020). However, until now it was not explored completely
how exceptional the year 2018 was in Germany and the Netherlands in terms of different drought indices.
This paper not only analyzes meteorological drought, but also agricultural drought with help of four



drought indices for Germany and the Netherlands. The drought analysis was based on HYDRUS 1-D
simulations using long-term observed climatic data as model forcing and using five different soil types
which are clay, silt, loam, loamy sand and sandy loam out of 12 textural soil classes. These five soil types
cover well the soil texture triangle and therefore the different types of soils. This modeling provides a
better understanding of drought trends and the 2018 drought event for 31 different locations over the
Netherlands and Germany with long-term meteorological observations.  In situ measured soil moisture
data and remotely sensed soil moisture are not available for such long time series and are in general
strongly affected by measurement uncertainties. Atmospheric reanalysis also provides soil moisture data,
but soil moisture is not an objective of such a reanalysis, and it is modelled at a relatively coarse scale and
with a simplified process representation. Therefore, simulated soil moisture content by a soil hydrological
model, driven by observed meteorological data, is expected to give the most reliable information on long
term (agricultural) drought trends. In this work, we analyzed calculated potential and actual
evapotranspiration (ET) as well as root zone soil moisture in detail. These three variables together with
precipitation were used to determine four drought indices: SPI, standardized soil moisture index (SSI),
potential precipitation deficit (PPD) and ET deficit. With the time series of the above-mentioned variables,
drought trends were analyzed, including the mechanisms behind the observed trends. In addition, it was
analyzed how extreme the 2018 drought event was in a historical context. The drought statistics and trends
were analyzed for different soil types, in order to determine whether drought trends were different for
the different soil types.
In Sect.2, we introduce the climatic stations over Germany and the Netherlands for which drought
assessments were made in this study, and also the model HYDRUS to simulate water flow in soils and
evapotranspiration is introduced. In addition, the study design and the four drought indices are introduced.
In Sect.3, we present the results of the drought trend analysis with a focus on extreme events and the year
2018. In Sect.4, we discuss and compare the drought trends and the extreme events with results from
other studies. Also limitations of this study are discussed. In Sect.5, we summarize this work with the most
important conclusions.

**2 Data and Methodology**
**2.1 Climatic data preparation**
In this study, 26 meteorological stations in Germany and 5 in the Netherlands were selected to investigate
drought trends over the period 1965-2019 and the uniqueness of the 2018 drought event in this region.
The stations are grouped in order to analyze domains with more similar climatological conditions:
Netherlands, northern Germany, western Germany, eastern Germany and southern Germany as shown in
the table 1.

**Table 1**. Analyzed meteorological stations and assignment to different domains.

| Domains | Sites |
| --- | --- |
| North Germany | Bremen, Bremerhaven, Hamburg, Hannover, Helgoland, List auf Sylt and Soltau |


| | |
|---|---|
| West Germany | Aachen, Essen, Gießen-Wettenberg, Koln-Bonn, Saarbrucken and Trier |
| East Germany | Artern, Chemnitz, Lindenberg, Potsdam, Magdeburg, Rostock-Warnemunde and Schwerin |
| South Germany | Augsburg, Freiburg, Hof, Nürnberg, Öhringen and Regensburg |
| Netherlands | Eelde, De Kooy, De Bilt, Vlissingen and Maastricht |


Figure 1 illustrates the spatial distribution of the meteorological stations. The meteorological stations are
located between 0 m above sea level (De Kooy) and 565 m above sea level (Hof, Germany). Data are from
the German Weather Service (DWD) (DWD Climate Data Center (CDC), 2021) and the Royal Dutch
Meteorological Institute (KNMI) (KNMI, 2021). Both weather services provide open-access climate data
including but not limited to precipitation, wind speed, relative humidity, maximum and minimum
temperature. Also incoming shortwave radiation is needed as forcing variable for the HYDRUS-1D version
4.17 model. The KNMI provides daily incoming shortwave radiation data for most of the meteorological
sites since the 1960s. The radiation data are also available for the DWD stations but for shorter time
periods and only for a few stations long term measurements are available. In this study, incoming
shortwave radiation was used in the simulations for the Dutch sites. For German sites, sunshine duration
was used to estimate incoming shortwave radiation according to the equation:
$R_s = (1 - \alpha)\left(a_s + b_s \frac{n}{N}\right) R_a$ (1)
where $R_s$ is incoming shortwave radiation [MJ m$^{-2}$d$^{-1}$], $R_a$ is extraterrestrial radiation [MJ m$^{-2}$d$^{-1}$], $\alpha$ is the
albedo or the canopy reflection coefficient (i.e., 0.23 for grassland), $a_s$ and $b_s$ are parameters for the
fraction of radiation (i.e., $a_s$ = 0.25, $b_s$ = 0.5), $n/N$ is the relative sunshine fraction [-], in which $n$ is the
number of measured sunshine hours and $N$ is the maximum possible number of sunshine hours for the
specific day of the year.



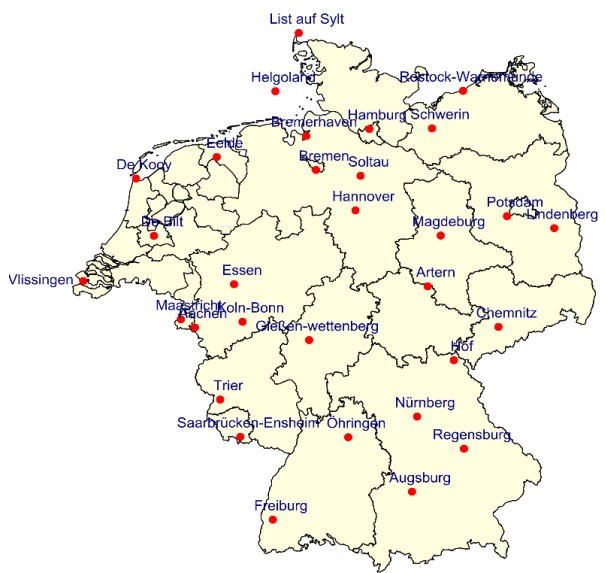


**Figure 1.** The 31 climatic stations used in this study, situated across Germany (26) and Netherlands (5) (data downloaded from DIVA-GIS [https://www.diva-gis.org/])

The four drought indices and drought events were calculated relative to a reference period of 55 years (1965 to 2019). This period has been selected so that a large number of stations could be considered for a relatively long period. However, not all stations provide complete datasets and some minor gaps exist in the datasets. The gaps were filled by a linear regression equation that was fitted using data from a neighboring station. Table 2 presents the neighboring stations used to fill data gaps by regression. Meteorological stations which are not listed in Table 2 do not have missing data.

**Table 2.** Meteorological stations and their neighboring stations, used to fill data gaps by linear regression

| Stations with gaps | Regression stations |
| --- | --- |
| Aachen | Köln-Bonn |
| Artern | Augsburg |
| Augsburg | Artern |
| Bremen | Bremerhaven |
| Bremerhaven | Bremen |
| Chemnitz | Hof |
| Essen | Köln-Bonn |
| Freiburg | Konstanz |
| Gießen-Wettenberg | Frankfurt-Main |
| Hamburg-Fuhlsbüttel | Soltau |
| Hannover | Soltau |
| Helgoland | List auf Sylt |
| Hof | Chemnitz |
| Köln-Bonn | Aachen |
| Lindenberg | Potsdam |



| | |
|---|---|
| List auf Sylt | Helgoland |
| Magdeburg | Gardelegen |
| Nürnberg | Regensburg |
| Öhringen | Freiburg |
| Potsdam | Lindenberg |
| Regensburg | Nürnberg |
| Rostock-Warnemünde | Schwerin |
| Saarbrücken-Ensheim | Trier-Petrisberg |
| Schwerin | Rostock-Warnemünde |
| Soltau | Bremen |
| Trier-Petrisberg | Saarbrücken-Ensheim |
| De Kooy | De Bilt |
| Eelde | De Bilt |
| Vlisingen | De Bilt |


**2.2 Methods**
**2.2.1 HYDRUS 1-D simulations**
Various models have been used to assess droughts (Mishra and Singh, 2011). In this study, the HYDRUS-
1D version 4.17 model (Simunek et al., 2005; Šimůnek et al., 2008; Huang et al., 2020) was applied to
calculate actual ET, soil moisture and resulting drought statistics. We simulated uniform water flow in 2
meters deep soil columns with homogeneous soil texture and a root uptake sink term.
All simulations are done for pasture with a rooting depth of 50cm. Flow for the 2m soil column is simulated
with the modified Richards equation:
$$\frac{\partial \theta}{\partial t} = \frac{\partial}{\partial z}\left[K(\theta)\left(\frac{\partial h}{\partial z} + 1\right)\right] - S + P - \mathrm{ET} \qquad (2)$$
where $\theta$ is volumetric water content [$L^3L^{-3}$], $t$ is time [T], $z$ is depth along the soil profile [L], $h$ is pressure
head [L], $S$ is source/sink term [$T^{-1}$], $P$ is precipitation [$T^{-1}$], ET is actual ET [$T^{-1}$], and $K(\vartheta)$ is the unsaturated
hydraulic conductivity function [$LT^{-1}$]:
$$K(\theta) = K_s\,K_r(\theta) \qquad (3)$$
where $K_s$ is saturated hydraulic conductivity [$LT^{-1}$] and $K_r$ the relative hydraulic conductivity [$LT^{-1}$]. The sink
term is the transpiration of the pasture here. $K_r$ is a function of soil moisture content (Van Genuchten,
1980):

$$K_r(\theta) = \theta^{1/2}\left[1 - (1 - \theta^{1/m})^m\right]^2 \qquad (4)$$
Unsaturated soil hydraulic properties were determined using the Van Genuchten model (Van Genuchten,
1980) with a fixed air-entry value of $-2cm$ in order to achieve better outcomes for finer soils:
$$\theta(h) = \begin{cases} \theta_r + \dfrac{\theta_s - \theta_r}{[1 + |\alpha h|^n]^m} & ,h < 0 \\ \theta_s & ,h > 0 \end{cases} \qquad (5)$$


$m = 1 - \frac{1}{n}, n > 1$                                                                    (6)
where $\alpha$ [L$^{-1}$] and $n$ are the inverse of the air-entry value and the pore-size distribution index respectively,
$\theta_s$ is saturated soil water content and $\theta_r$ is residual soil water content. The lower boundary is a free
drainage boundary condition and the upper boundary is governed by atmospheric conditions without
surface runoff and ponding. The potential ET was calculated using the Penman-Monteith equation (Allen
et al., 1998) with the assumption of a pasture height of 12 cm, an albedo of 0.23, a LAI of 2.0 and a rooting
depth of 50 cm. Alternatively, simulations were also performed with a higher LAI-value of 2.88, to
investigate the impact of LAI on the results.

The models were run for five soil types (clay, silt, loam, loamy sand and sandy loam) in order to see the
impact of soil type on the drought indices. The soil hydraulic properties were determined from texture
according to Carsel and Parrish (1988), see also Table 3.

**Table 3.** Soil hydraulic parameter values for the selected texture types in the simulations

| Parameters | Clay | Silt | Loam | Sandy loam | Loamy sand |
|---|---|---|---|---|---|
| $K_s$, cm d$^{-1}$ | 4.8 | 6 | 24.96 | 106.1 | 350.2 |
| $\alpha$, cm$^{-1}$ | 0.008 | 0.016 | 0.036 | 0.075 | 0.124 |
| n | 1.09 | 1.37 | 1.56 | 1.89 | 2.28 |
| $\theta_s$ | 0.38 | 0.46 | 0.43 | 0.41 | 0.41 |
| $\theta_r$ | 0.068 | 0.034 | 0.078 | 0.065 | 0.057 |


We monitored soil moisture in the simulated soil columns at five different soil depths and calculated
average soil water content on the basis of these five values. These monitoring schemes differed slightly
between soil types. For the fine soils (clay, silt and loam) the vertical discretization was 2.5mm and
monitoring nodes were inserted at 4.75 cm, 15.5 cm, 25.25 cm, 35.25 cm and 45.25 cm, covering the root
zone of 50cm. For the coarse soils (sandy loam and loamy sand) the vertical discretization was 2.0mm and
monitoring nodes were inserted at 4.8 cm, 15 cm, 25 cm, 35cm and 51cm depth.
**2.2.2 Drought indices**
Due to the complex nature of droughts, one single drought index is not sufficient for drought detection
(Hao and AghaKouchak, 2014). In this study, we focus on meteorological drought and agricultural drought.
In this context, four drought indices are evaluated.
**2.2.2.1 The Standardized Precipitation Index (SPI)**
The SPI was developed by (McKee et al., 1993) and is calculated based on long-term precipitation data
(minimum 30 years). A time series of precipitation data is transformed to a normal distribution after
fitting to a probability distribution (McKee et al., 1993), which allows for estimation of dry and wet
periods. Positive SPI values greater than median precipitation means wetter conditions than normal. The



SPI can be calculated for different time scales (i.e. 3, 6, 9, 12 and 24 months) and can be calculated as
follows:
$$\text{SPI}_{i,j} = \frac{\text{P}_{i,j} - \overline{\text{P}}_j}{\sigma_{P,j}}$$ (7)
where $\text{P}_{i,j}$ [mm] is the precipitation for time interval $j$ (6 months or 12 months in this study) of the year $i$,
$\overline{\text{P}}_j$ [mm] is the average precipitation over the studied period and $\sigma_{P,j}$ [mm] is the standard deviation of
precipitation for time interval $j$ over the studied period (Cavus and Aksoy, 2019; Fathian et al., 2021).
Extreme droughts are characterized by negative SPI values, see Table 4. SPI is one of the most used
drought indices (Mishra and Singh, 2010; Gudmundsson and Seneviratne, 2015; Zink et al., 2016). It is
also a drought index recommended by the World Meteorological Organization (WMO) and German
Weather Service (DWD). Compared to another widely used drought index, the Palmer drought severity
index (PDSI) (Palmer, 1965), SPI has the advantage of simplicity because it only requires information on
precipitation. SPI has limitations for shorter time series and dry climates, but in this study long time
series of 55 years (1965 – 2019) are calculated and the climate over the study domain is in general
temperate humid, although drier in Eastern Germany. The severity of droughts over the simulation time
period has been assessed for the summer period from April to September (SPI-6), as well as for periods
of 12 months (SPI-12). Droughts are classified into four classes, ranging from mild droughts to extreme
droughts (McKee et al., 1993). See also Table 4.
**Table 4.** Classification of SPI

| SPI-values | Drought classification |
| --- | --- |
| Between -0.99 and 0 | Mild drought |
| Between -1.49 and -1.00 | Moderate drought |
| Between -1.99 and -1.50 | Severe drought |
| Less than – 2.00 | Extreme drought |


**2.2.2.2 Standardized Soil Moisture Index (SSI)**
Similar to SPI, SSI is also calculated by deriving probabilities from soil moisture time series (AghaKouchak,
2014; Hao and AghaKouchak, 2014), which allows to determine periods with negative and positive soil
moisture anomalies. SSI is calculated in this work for two time intervals, SSI-6 for summer droughts (April-
September) and SSI-12 for annual droughts. SSI is therefore determined using the same principles as SPI.
Therefore, the SSI can be calculated as follows:
$$\text{SSI}_{i,j} = \frac{\theta_{i,j} - \overline{\theta}_j}{\sigma_{\theta,j}}$$ (8)
Where $\theta_{i,j}$ [mm] is the soil moisture content for time interval $j$ (6 months or 12 months in this study) of
the year $i$, $\overline{\theta}_j$ [mm] is the average soil moisture content over the studied period and $\sigma_{\theta,j}$ [mm] is the
standard deviation of soil moisture content for time interval $j$ over the studied period.



**229**  **2.2.2.3 Potential Precipitation Deficit (PPD)**

**230** Potential precipitation deficit is a drought index developed by the Royal Dutch Meteorological Institute
**231** (KNMI) to assess drought events and it is defined by the difference between potential ET and precipitation.
**232** PPD also stands for meteorological drought but in addition to SPI, PPD includes information on the
**233** atmospheric evapotranspiration demand. Specific for the conditions in the Netherlands and Germany, the
**234** deficit is calculated for the period between April and September with large potential ET. PPD is defined as
**235** the cumulative deficit between potential ET according to the Penman Monteith method and precipitation
**236** and calculated for each day in the period between the 1$^{st}$ of April and 30$^{th}$ of September. Typically, PPD
**237** increases during summer as potential ET is larger than precipitation. PPD cannot become negative (if
**238** cumulative precipitation is larger than cumulative potential ET) and will have as minimum value zero. The
**239** maximum PPD value reached in the period between the 1$^{st}$ of April and the 30$^{th}$ of September (typically in
**240** August) is recorded.

**241**  **2.2.2.4 Evapotranspiration Deficit (ET$_{def}$)**

**242** A direct indicator for drought stress affecting vegetation is the difference between potential ET and actual
**243** ET. If actual ET is smaller than potential ET, this indicates that not enough soil water was available to satisfy
**244** the atmospheric demand. This also implies that crop yield is affected. We define therefore as further
**245** indicator for drought stress (ET$_{def}$):

**246**
$$\mathrm{ET_{def}} = \sum_{i=1}^{n}\left(\mathrm{ET_{pot,i}} - \mathrm{ET_{act,i}}\right) \tag{9}$$

**247** where ET$_{pot}$ is potential ET, ET$_{act}$ is actual ET, $i$ is a day between the 1$^{st}$ of April ($i$=1) and the 30$^{th}$ of
**248** September ($i$=183) and $n$ is 183 days.

**249**  **2.2.3 Trend analysis**

**250** Trends are analyzed in this study according the Theil-Sen slope (Theil, 1950; Sen, 1968) and the significance
**251** is assessed based on the Mann-Kendall (MK) test with p value 0.05, which allows for pre-whitening in order
**252** to eliminate the impact of autocorrelation (Hamed and Rao, 1998; Yue and Wang, 2004). The MK test is
**253** nonparametric and is regularly used in hydrometeorological trend detection and the python package
**254** 'pymannkendall' is used for computations.

**255**

**256**  **3 Results**

**257**  **3.1 Trends of precipitation, potential ET, actual ET, soil moisture and drought indices for 1965-2019**

**258**  **3.1.1 Precipitation trends**

**259** Figure 2 displays trends of precipitation for different regions in Germany and the Netherlands. Coinciding
**260** with Spinoni's study (Spinoni et al., 2015), Fig. 2 shows that precipitation for all four areas of Germany
**261** show decreasing trends, but with clear differences. In southern Germany, the precipitation decrease was
**262** strongest (-2.2 mm/year) while in the East the decreasing trend was only -0.23 mm/year. Different to the
**263** German regions, the Netherlands experienced a small increase in precipitation at the rate of 0.385
**264** mm/year. Spinoni et al. (2015) reported also an increase for the Netherlands, but for the period 1950 –
**265** 2012.






(a) (b)

(c) (d)

(e)





**Figure 2.** Precipitation trends in (a) North Germany, (b) West Germany, (c) East Germany, (d) South Germany and (e) Netherlands. Shaded areas indicate the 95% confidence interval estimated from (the limited) number of sites in each domain.

### 3.1.2 Potential ET and actual ET trends

Figure 3 shows that potential ET has increased in all areas, with trends varying between 1.04 mm/year (northern Germany) and 1.66 mm/year (Netherlands). Table 5 also shows the potential ET trends, and indicates that for all five regions considered here the trend of increasing potential ET is significant (coinciding with Spinino et al. (2015)). However, actual ET trends show different trend signs (Fig. 3), with increasing trends for Northern Germany, Eastern Germany and Netherlands and decreasing trends for Western Germany and Southern Germany. The decreasing trends in the West and South of Germany can be related to strong decreasing precipitation trends in those two domains.

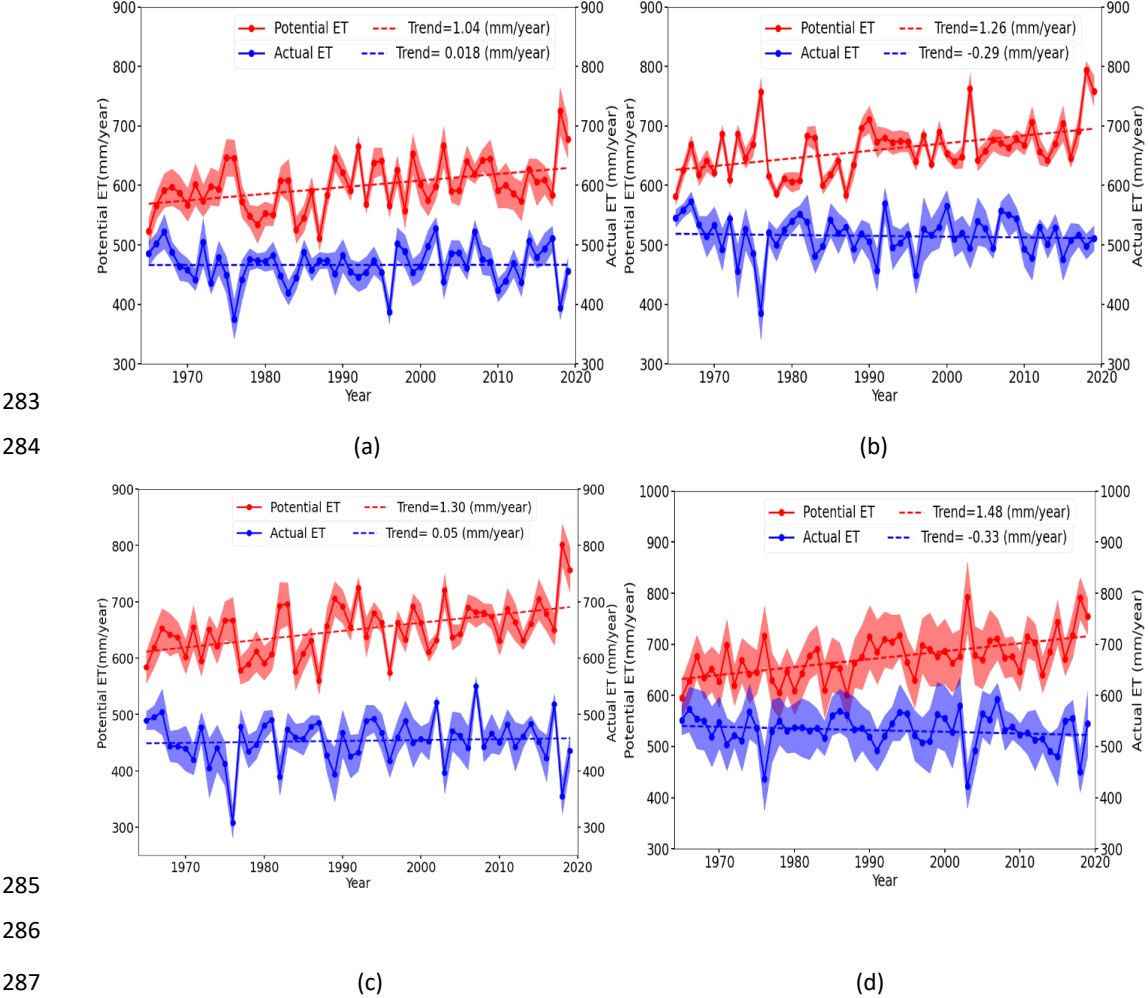

(a)

(b)

(c)

(d)





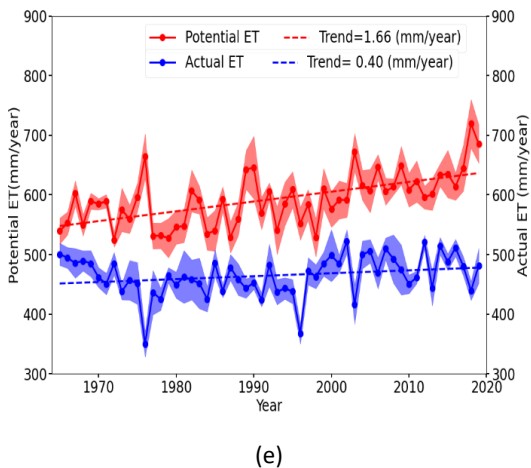


(e)

**Figure 3.** Potential ET (red line) and actual ET (blue line) trends in the (a) North, (b) West, (c) East and (d)
South of Germany and (e) Netherlands. Shaded areas indicate the 95% confidence interval estimated
from (the limited) number of sites in each domain.


**3.1.3 Soil moisture trends**
Figure 4 displays the temporal trend in the average yearly soil moisture content, for the four German
domains, the Netherlands and the five different soils. As expected, soil water contents are highest for clay
and lowest for sandy loam and loamy sand. For all soils a decreasing soil moisture trend can be observed
over all domains, which is significant for clay and silt, but not significant for all the other soil types. The
trends are less pronounced for the Netherlands and eastern Germany than for the other three domains.
For all soil types the year 2018 has the lowest average soil moisture content except for 1976 in western
Germany and Netherlands. The different soil types show a similar ranking of the different years but
interannual variability is higher for loam and silt.

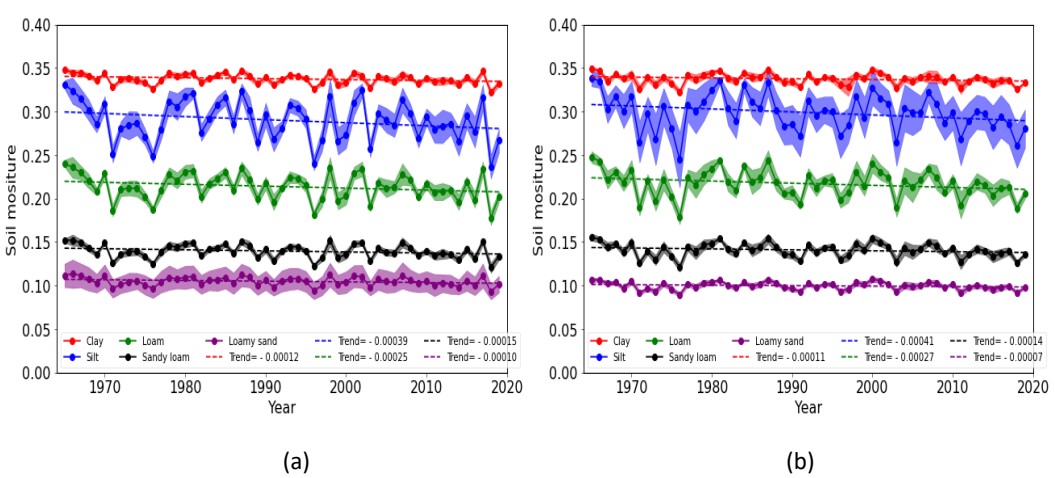


(a)                                              (b)



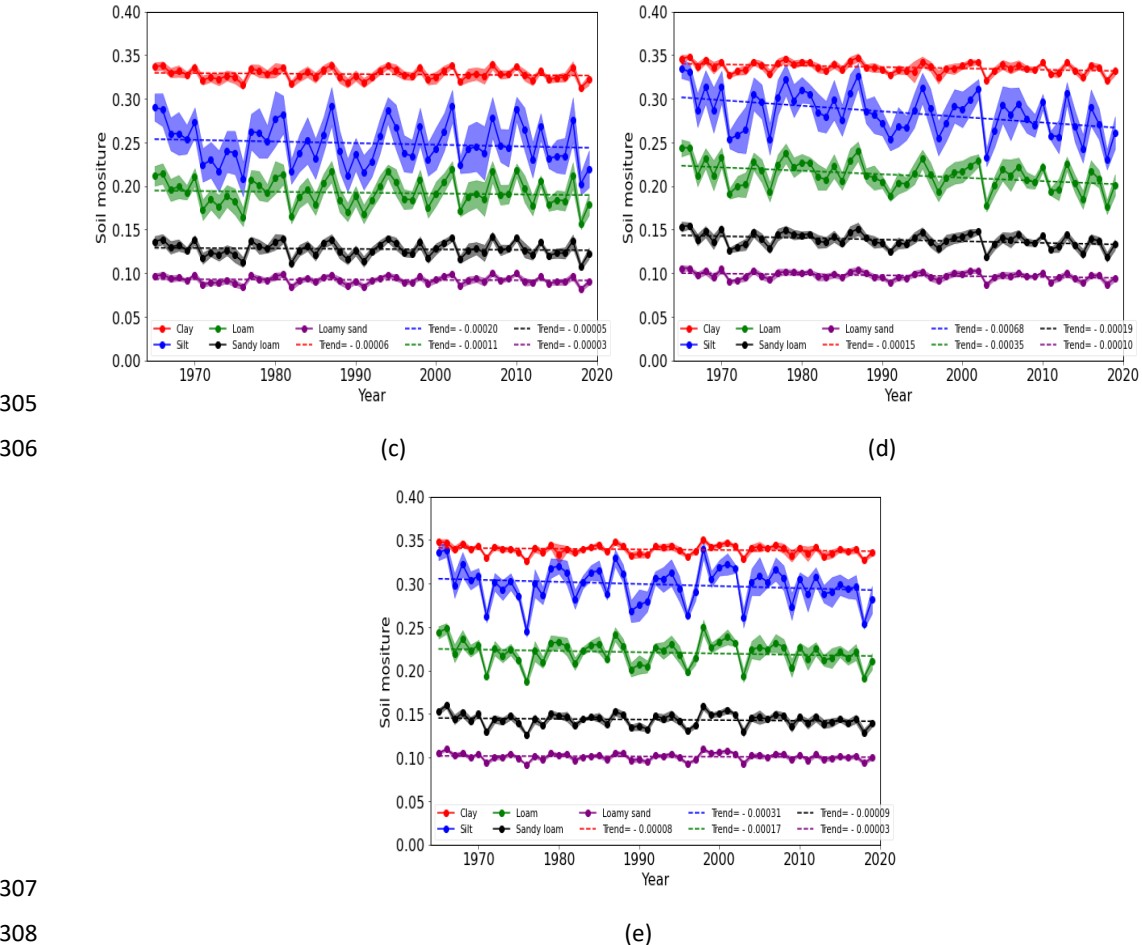


(c)                               (d)


(e)

**Figure 4.** Trends in average yearly soil moisture content for (a) North, (b) West, (c) East and (d) South of
Germany and (e) the Netherlands, for five different soil textures (clay (red), silt (blue), loam (green),
sandy loam (black) and purple (loamy sand)). The trend coefficients are also indicated in each of the
figures. Shaded areas indicate the 95% confidence interval estimated from (the limited) number of sites
in each domain.

Table 5 shows the trends in potential ET, actual ET and soil moisture content simulated by HYDRUS-1D
version 4.17, for the different regions in Germany and the Netherlands. Actual ET shows a non-significant
decreasing trend in western and southern Germany which can be explained by decreasing precipitation.
The Netherlands shows the strongest increase in actual ET, which can be explained by the increasing
precipitation and potential ET trends. In northern and eastern Germany, the actual ET trends show a slight
non-significant increase, in spite of the decreasing precipitation trend, which can be explained by the fact
that actual ET is often energy limited, and not moisture limited. Root zone soil moisture content as
simulated by HYDRUS shows over all regions a non-significant decreasing trend, except for southern





Germany where the trend is significant. The decreasing trends can be explained by a combination of
decreasing precipitation (in most regions) and increasing potential ET.

**Table 5.** Trends for potential ET, actual ET and soil moisture for the different studied regions in Germany
and the Netherlands over the period 1965-2019

| Regions | Potential ET (mm/year) | Significance | Actual ET (mm/year) | Significance | Soil moisture ($10^{-3}m^3m^{-3}$) | Significance |
|---------|------------------------|--------------|---------------------|--------------|-------------------------------------|--------------|
| Germany-East | +1.30 | yes | +0.05 | no | -0.09 | no |
| Germany-West | +1.25 | yes | -0.28 | no | -0.21 | no |
| Germany-North | +1.04 | yes | +0.018 | no | -0.20 | no |
| Germany-South | +1.48 | yes | -0.33 | no | -0.29 | yes |
| Netherlands | +1.66 | yes | +0.40 | no | -0.13 | no |


### 3.1.4 Temporal trends of SPI, SSI, PPD and ET deficit

The four drought indices, determined for the 31 meteorological stations, were averaged for the five
domains and the yearly values and trends are displayed in Figs. 5 and 6. The SPI shows decreasing trends
over all four German regions, which is indicative of increased frequency of meteorological drought
conditions. The decreasing trend is strongest over southern and western Germany, and only significant for
southern Germany. On the contrary, for the Netherlands an increasing trend can be observed. All five
regions show a decreasing SSI-trend, in this case also the Netherlands. The negative trends indicate
increasing frequency of agricultural droughts. For all five regions the slope of the SSI-regression line is
more negative than the slope of the SPI-regression line, which indicates that the frequency and/or
intensity of agricultural droughts is increasing faster than the frequency and/or intensity of meteorological
droughts. This indicates that besides a decrease in precipitation, there are factors which enhance
agricultural droughts. Earlier we saw that for all regions in Germany and the Netherlands a significant
increase of potential ET over the period 1965-2019 was detected. The negative SSI-trends are strongest
for southern and western Germany.
The potential precipitation deficit (PPD) and ET deficit show positive trends for all five regions, pointing
towards more agricultural drought, as water is lacking to fulfill the potential evapotranspiration demand.
In western and southern Germany PPD and ET deficit show trends between 1.5 and 1.7 mm/year, which
is larger than for the Netherlands, northern and eastern Germany (1.0-1.2 mm/year). This is related to the
strong and significant increase of potential ET over time, in combination with only small changes in
precipitation amount. The trends are in all cases significant, except PPD in northern and eastern Germany.





(a)



(b)



(c)

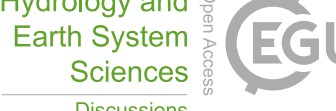

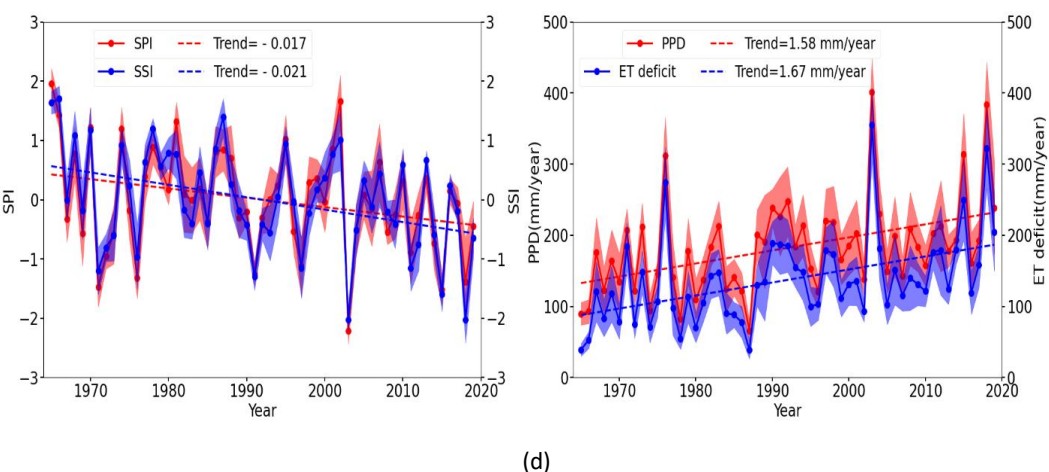


(d)

**Figure 5.** Temporal trends (1965-2019) for SPI and SSI (left), and PPD and ET deficit (right) in (a) northern
Germany, (b) western Germany, (c) eastern Germany and (d) southern Germany. Shaded areas indicate
the 95% confidence interval estimated from (the limited) number of sites in each domain.

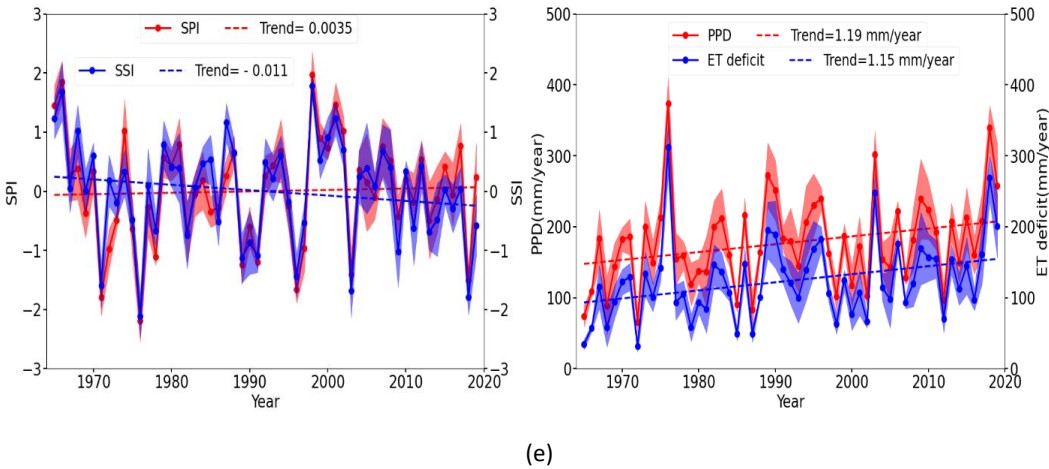


(e)

**Figure 6.** Temporal trends (1965-2019) for SPI and SSI (left), and PPD and ET deficit (right) over the
Netherlands. Shaded areas indicate the 95% confidence interval estimated from (the limited) number of
sites in each domain.


We analyze now the SPI, SSI, PPD and ET deficit trends for the 31 individual stations. Table 6 summarizes
results from a trend analysis of the drought indices for the summer period from the 1st of April until the
30th of September. The SPI-6 decreased for 23 out of 31 stations (and increased for the other 8 stations)
with a mean trend of -0.071 decade[-1] averaged over all 31 stations, only for 4 stations this decrease was
significant ($p < 0.05$). Results are very similar for SPI-12, but for two more sites the decreasing trend was
significant. Gudmundsson and Seneviratne (2015) also calculated SPI-12 at the scale of Europe but their
results were not provided for individual countries or regions. They detected non-significant trends of
different signs over Germany.



Summer soil moisture drought, as evaluated by SSI-6, shows a clear trend of -0.186 decade$^{-1}$ averaged over
all 31 stations. All stations show a negative trend, and for 17 of these stations there is a significant negative
trend. On the other hand, soil moisture droughts at the yearly time scale, as evaluated by SSI-12, do not
show such a pronounced negative trend and only for 8 stations a significant negative trend is found. For 7
stations even a trend towards increased SSI-12 values is observed. Five of these seven stations are located
in the Netherlands, again indicating that the Netherlands is less impacted by a trend towards increasing
droughts than Germany.
Finally, both PPD and ET deficit show positive trends indicating increased droughts with an increase of PPD
of 1.25 mm/year and increase of ET deficit of 1.30 mm/year averaged over all stations. For PPD 15 of the
stations and for ET deficit 20 of the stations show significant trends ($p<0.05$).
In summary, the increase of droughts is mainly related to increasing soil moisture deficits, and reduction
in actual ET. The main driver is not a precipitation decrease, but an increase of potential ET.

**Table 6.** Number of stations with increasing drought trends and in brackets the number of significant
drought trends. The mean trend is also given.

| Drought indices | SPI-6 | SPI-12 | SSI-6 | SSI-12 | PPD (mm) | ET deficit (mm) |
|---|---|---|---|---|---|---|
| Numbers of sites | 23 (4) | 22 (6) | 31 (17) | 24 (8) | 31 (15) | 31 (20) |
| Mean trend (year$^{-1}$) | -0.0071 | -0.0068 | -0.0186 | -0.0004 | 1.25 | 1.30 |



**3.2 The year 2018 and other drought years during 1965-2019**
**3.2.1 Climatological variables for 2018 in a historical perspective**
We ranked the position of the year 2018 in the time series of 1965-2019 for six climatic variables and for
all 31 meteorological stations. Figure 7 shows for each of the six variables for how many of the 31
meteorological stations 2018 ranked in the top three (Tier 1), in the top ten but not in the top three (Tier
2), and outside the top ten (Tier 3). Top-ranking implies that meteorological variables were prone to result
in drought conditions: high temperatures, high incoming shortwave radiation, low precipitation, low
relative humidity and high wind speed. Figure 7 shows that for 30 out of 31 stations the average maximum
temperature for 2018 ranks in the top three. Also the total amount of sunshine hours was very high for
2018 with a top three position for 28 out of 31 stations. Rankings are less pronounced for low relative
humidity (top three for 23 out of 31 stations), low precipitation (for 14 out of 31 stations in the top three
of driest years). For daily average minimum temperature, the year 2018 showed less extreme rankings
(but often in top ten), and wind speed deviated for most stations not much from a normal year (outside
top ten).

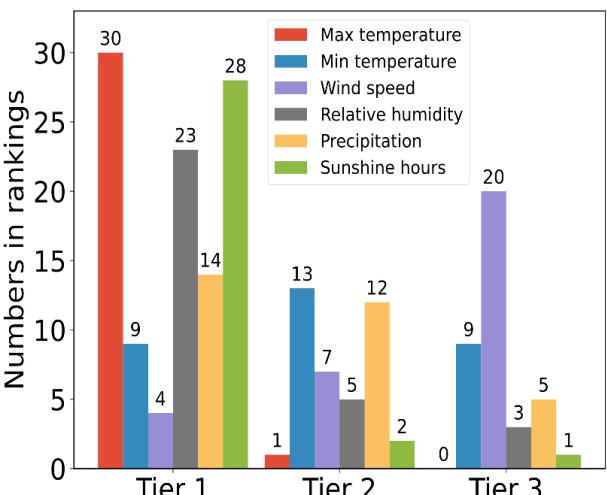


**Figure 7.** Ranking of climatic variables for the year 2018 in the time series of 1965-2019 (Tier 1=Ranks 1-3, Tier 2=ranks 4-10 and Tier 3 =ranks 11-31)


### 3.2.2 Potential ET and actual ET extremes

Figure 8 displays the years with the highest potential ET and the lowest actual ET for the 31 meteorological
stations. Except two stations in southern Germany (Augsburg and Freiburg), 2018 is the year with the
highest potential ET. Augsburg and Freiburg had in 2003 the highest potential ET. The summer of 2003 was
extremely warm over Central Europe (especially Southern Germany, Eastern France, Switzerland and
Austria). Regarding actual ET, at some meteorological stations (according calculations with HYDRUS-1D
version 4.17) 2018 had the lowest total sum of actual ET over the complete time series from 1965-2019,
this is the case for four sites across Germany. However, the year 1976 with the historic summer drought
had for many meteorological stations the lowest actual ET, due to a combination of low precipitation and
high atmospheric evaporative demand. This was especially the case for northern and central Germany.
Three sites in Germany had the lowest calculated actual ET in 2003 and two Dutch stations (Eelde and De
Bilt) had the lowest calculated actual ET in 1996, which was related to a combination of a relatively limited
potential ET and drought.



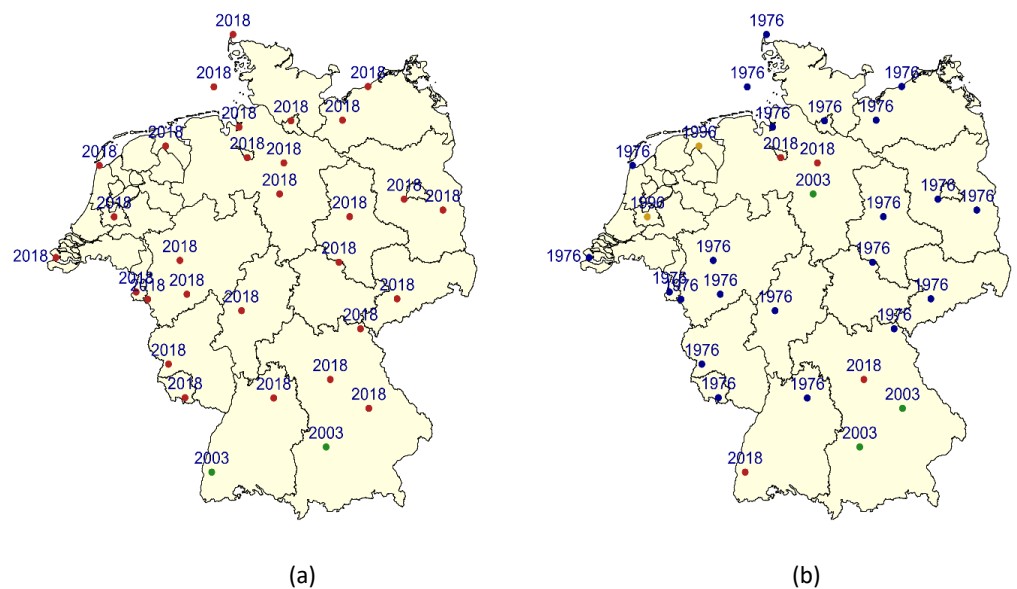

(a)                                              (b)

**Figure 8.** The years with the (a) highest potential ET according Penman-Monteith and (b) lowest actual ET (HYDRUS-calculations) at the different 31 meteorological stations across the Netherlands and Germany (data downloaded from DIVA-GIS [https://www.diva-gis.org/])

### 3.2.3 The most extreme drought events based on drought indices

Figure 9 shows the year with the lowest SPI (precipitation deficit), the lowest SSI (soil moisture deficit), the largest potential precipitation deficit and the largest ET deficit. The years 1976, 2003 and 2018, all of them with very warm and dry summers, are most prominent on the maps. For SPI, the year 1976 dominates over the southern Netherlands and parts of western Germany, 2003 over southern Germany and 2018 over other parts of Germany. For SPI, also other years like 1971 and 1996 appear on the map. It indicates that meteorological droughts are spatially variable over the past 55 years. For SSI, the spatial patterns are not very different from SPI but 2018 is at more stations the year that results in the lowest SSI. If we analyze now the years with the most extreme PPD and ET deficits, nearly exclusively 1976, 2003 and 2018 are the extreme years. One exception is the 1989 drought in the state Saxony-Anhalt in Germany. The year 2003 has in general the highest PPD and ET deficits in southern Germany, the year 1976 in the Netherlands and western Germany and the year 2018 in northern and eastern Germany. In summary, the drought event 2018 impacted especially northern and eastern Germany and is characterized by very high potential ET (besides lower precipitation), while the 1976 drought is in general characterized by high potential ET (but not as high as in 2018) and very low precipitation (often lower than in 2018).







**Figure 9.** The most extreme drought years according different drought indices: (a) SPI; (b) SSI; (c) PPD and (d) ET Deficit (data downloaded from DIVA-GIS [https://www.diva-gis.org/])

We analyze further the three most extreme drought events (1976, 2003, 2018) over the region. Table 7 summarizes averaged drought indices over the complete domain (Netherlands and Germany). According to the SPI-6 index the 1976 drought can be considered as an extreme drought, whereas the droughts in 2003 and 2018 are classified as severe droughts. Also the other meteorological drought index, SPI-12, has the most extreme value for the year 1976. Concerning soil moisture droughts, SSI-6 ranks first for 1976, but SSI-12 ranks first for 2018. This is related to the fact that 2018 also had a (very) dry spring and autumn in large parts of the Netherlands and Germany. Both for SSI-6 and SSI-12 the year 2003 ranks 3rd. For the



drought indices which include potential ET (PPD and ET deficit), the summer of 2018 ranks first and before
1976 and 2003; 2018 is more extreme than 1976 (2nd) and 2003 (3rd). These indices show again that the
2018 drought was extreme, especially related to the very high potential ET and the lack of water for plant
transpiration.

**Table 7.** Drought severity in 2018 in comparison with 1976 and 2003. The most extreme values are
printed bold italic.

| Year | SPI-6 | SPI-12 | SSI-6 | SSI-12 | PPD (mm) | ET deficit (mm) |
|------|-------|--------|-------|--------|----------|-----------------|
| 2018 | -1.71 | -1.56 | -1.60 | *-1.92* | *399.6* | *329.2* |
| 2003 | -1.22 | -1.47 | -1.62 | -1.52 | 327.0 | 279.8 |
| 1976 | *-2.11* | *-1.81* | *-1.87* | -1.66 | 368.4 | 313.9 |

**3.3 Simulations for different LAI**
Alternatively, we performed simulations with a higher LAI-value equal to 2.88, to investigate the impact
on the results. Potential ET is strongly related to LAI, and is larger for higher LAI. Figure S1 (supplementary
material) shows that potential ET is always considerably larger for a LAI of 2.88 compared to a LAI of 2.0,
50 to 100mm on year basis, and also the trends are slightly stronger.
Although LAI impacts potential (and actual) ET, its impact on yearly average soil moisture content is limited,
and simulated soil moisture contents for LAI equal to 2.88 are only very slightly smaller than soil moisture
contents for LAI equal to 2.0. The soil moisture trends are also nearly identical for the two different LAI-
values. Plots are shown in the supplementary material (Fig. S2).
Figures S3 (supplementary material) and S4 (supplementary material) show the PPD and ET-deficit trends
for the different regions and for the two different LAI-values. The figures illustrate that LAI=2.88 results in
higher PPD and higher ET-deficit than LAI=2.0. For PPD, the largest differences between the temporal
trends for LAI=2.0 and LAI=2.88 are simulated for northern Germany where the PPD-trend is 0.99 mm/year
for LAI=2.0 and 1.36 mm/year for LAI=2.88. For ET-deficit, western Germany, southern Germany and
northern Germany show the largest increase in the slope of the trend line, comparing LAI=2.88 and LAI=2.0.
However, the overall picture is the same for LAI=2.0 and LAI=2.88, and the main difference is the
magnitude of PPD and ET-deficit.
**4 Discussion**
This study was carried out for pasture, and we investigated also the impact of different LAI-values on the
simulated temporal trends in drought indices. However, for other vegetation types with deeper roots and
higher LAI, the temporal trends and rankings we found here could be different. It can be expected that
deeper rooting vegetation like forests can take up water from deeper soil layers so that forests are not
affected so fast by droughts as crops or pasture (Kleine et al., 2020). As a consequence, the ET-deficit might
show a less pronounced temporal trend for forests. It is expected that the other drought indices (SPI, SSI,
PPD) for forests are quite similar to grassland.



In our simulations, we used a free drainage lower boundary condition. Alternatively, the lower boundary
condition could be a groundwater table, whose level typically also varies over time and is lower during
summer time and shallower during winter time. The simulated temporal trends of drought indices could
be affected by the lower boundary condition, as groundwater affects soil moisture contents and actual
evapotranspiration. During drought conditions, it can provide an additional water source for the
vegetation and buffer droughts, resulting in a smaller ET-deficit. The presence of a shallow groundwater
body can therefore also affect the ranking of the most extreme drought years. For example, if a drought
year is preceded by a wet winter which resulted in relatively high groundwater tables at the beginning of
the summer season, the summer drought can be buffered by the relatively high groundwater tables,
reducing the ET-deficit compared to a situation with deeper groundwater tables. A further limitation is
that we assumed a no ponding condition at the land surface. This assumption can also affect the simulated
removal of water from the soil column, soil moisture contents and actual evapotranspiration.

## 5 Conclusions

We analyzed meteorological, soil moisture and agricultural droughts over Germany and the Netherlands
for the period 1965-2019 with separate analysis for four different German regions (north, west, east and
south). It was evaluated how exceptional the 2018 drought was. Simulations were done for 31 locations
distributed over Germany and the Netherlands with long meteorological time series to drive the HYDRUS-
1D model for unsaturated flow in soils. Simulations were done for five different soil types at each location
and for pasture with two different LAI-values. The main conclusions are:

1.  The year 2018 experienced very high maximum temperatures, very high sunshine duration, and
    low relative humidity across the study domain, and was for most studied sites the year with the
    highest potential ET. In terms of precipitation deficit, 2018 was for many locations not exceptional
    and only for 8 locations (out of 31) the standardized precipitation index (SPI) had in 2018 the
    lowest value in the time series of 1965-2019. The years 1976 (especially in the Netherlands and
    western Germany) and 2003 (especially in southern Germany) were often drier in terms of SPI.
    The year 2018 was more exceptional in terms of soil moisture drought and for 13 out of 31 stations
    SSI (standardized soil moisture index) was in 2018 the lowest in the time series of 55 years. This
    was especially the case for northern and eastern Germany. The year 2018 was remarkable in terms
    of potential precipitation deficit (PPD, precipitation minus potential ET) and evapotranspiration
    deficit (accumulated difference between potential ET and actual ET), related to the very high
    potential ET. For half of the stations (especially in northern, central and eastern Germany) 2018
    was the most extreme year in the time series since 1965. This illustrates that the 2018 drought
    was especially extreme and intensified by the high potential ET, affecting reductions in crop yield
    due to the missing water for plant transpiration. Other years with wide spread severe droughts
    were especially 1976 and 2003, but in terms of soil moisture drought and ET drought they were
    less extreme than 2018 over the study domain.

2.  The trends in the four drought indices over the period 1965-2019 showed that meteorological
    droughts (in terms of SPI-6 and SPI-12) increased at many locations, especially in Germany,
    whereas at other locations meteorological drought decreased, especially in the Netherlands. Only
    at a few locations the trend towards more severe meteorological droughts were significant. Soil
    moisture or agricultural drought, as characterized by SSI-6 and SSI-12 increases at most locations,




SSI-6 even at all locations. SSI-6 shows a significant trend towards more soil moisture droughts for
slightly more than half of the locations. PPD and ET deficit show even more pronounced trends
with trends towards increased water deficit at all locations, and significant trends for 15 out of 31
(PPD) and 20 out of 31 (ET deficit) locations. Significant trends towards more severe droughts in
terms of SSI, PPD and ET deficit occur in spite of small changes in total precipitation amounts.
These significant trends are driven by increased potential ET (related to higher temperatures,
higher incoming radiation and lower relative humidity) and a shift of precipitation amounts during
the year with increasing precipitation in winter and decreasing precipitation in summer.
3.  Simulations were limited to pasture, a relatively simple representation of vegetation, and a free
drainage boundary condition. It would be of interest to extend the simulations to include different
crop types and natural vegetation, as well as different boundary conditions including
groundwater-controlled systems. We expect for crops similar results as for pasture, but for deep
rooting vegetation and groundwater-controlled systems the ranking of the different drought years
might change, for example if at the beginning of a drought year groundwater levels are relatively
high for the time of year, allowing for an additional water buffer during the dry summer.

**Data and code availability**

The code and datasets relevant to this study can be provided from the corresponding author upon request.

**Author contributions**

HJHF, YH and JW designed the study. YH (completely) and JW (partially) developed the programming code
and performed the simulations. YH, JW and HJHF analysed and interpreted the results. HJHF and HV
supervised the work. YH prepared the manuscript with contributions from all co-authors.

**Competing interests**

The authors declare no competing interests.

**Acknowledgement**

Yafei Huang (No. 201608500073) appreciates financial support from the China Scholarship Council.
Harrie-Jan Hendricks-Franssen kindly acknowledges support from Germany's Excellence Strategy (EXC
2070–390732324, project PhenoRob).

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
