# Peer review of "Long-term trends in agricultural droughts over Netherlands and Germany: how extreme was the year 2018?"

_Hydrology and Earth System Sciences, 2021_

## Author Comment (AC2)

**Response (in blue) to comments by Anonymous Referee #1**

*This study aims to assess how extreme the drought year 2018 was in Germany and the Netherlands, based on standard drought indices (SPI, SSI), also potential precipitation deficit (PPD), and ET deficit. The study used HYDRUS 1D simulations for 31 stations with long-term meteo observations, and calculated soil moisture, potential ET, and actual ET for five soil types. Their results show that the increasing droughts over Germany and the Netherlands are mainly driven by increasing potential ET and increasing vegetation water demand. While the topic is relevant and analysis is interesting, this reviewer found the manuscript cannot be accepted with the current form.*

Response: We very much appreciate the comments by the reviewer. We will revise the manuscript taking into account the comments.

*Major concern:*
*This reviewer found that the study/experiment design could be flawed, mainly due to the lack of description on how representative the five soil types used for representing the different domains, and the lack of description on why the pasture is assumed for all these stations. To this reviewer, the current study is merely a synthetic study. Thus, it is far from understanding the drought year 2018 over the Netherlands and Germany.*

Response: We thank the reviewer for this comment and the opportunity to clarify better our objectives. We will in the revised version of the paper also improve the clarification of our objectives, taking into account the reviewer comments.

The five soil types are representative for the domain because they cover well the soil texture triangle and for each location calculations are repeated for these five soil types, covering different possible conditions near the measurement sites. It can be expected that in a region around a measurement site all these different soil types are present. In this study, our main objective is not to determine for each location as good as possible what soil moisture and evapotranspiration conditions were in 2018. This would require the use of all possible information sources including remote sensing information on soil moisture and vegetation states, precise soil and land use land cover information, among others. Our objective was to make a standardized comparison with past years and past droughts. For the further past, especially before the year 2000, remote sensing information is not of good quality, and a data-based comparison is not possible. This is the reason why we used a model-based comparison, covering all possible soil types.

The pasture was chosen because in this case it is known that all 31 meteorological stations are default located on a pasture. However, we agree with the reviewer that in a similar manner as for soil types, it would have been possible to cover different vegetation types (e.g., grassland, crop land, forest). To some limited extent, we took different vegetation states into account by performing calculations with two different LAI time series. We decided not to include simulations for different vegetation types given the large amount of information already

included in the paper, and also because we found that for the five different soil types, the drought trends and ranking of the drought years were hardly affected by soil type, in spite of the fact that absolute soil moisture contents showed large differences between soil types. For the two LAI time series smaller absolute differences among sites were observed (than for soil types), and the ranking was not affected. We already included a discussion on the impact of vegetation type on the results, pointing to the fact that for deep rooting vegetation rankings could be affected. We consider that our model-based comparison already covers many different conditions and think that a further extension is beyond the scope of this manuscript, and will not affect the main conclusions of this paper.

We will modify the manuscript to better motivate why we used this study design, and that the main objective is a standardized model-based comparison in the context of a lack of precise data for the years further in the past. We will stress that the objective is not to make site-specific best estimates of soil moisture and ET conditions in the past.

---

## Author Comment (AC3)

**Response (in blue) to comments by Anonymous Referee #2**

*The authors use Hydrus-1D to simulate water balance to test the impact of meteorological drought on the agricultural drought over 31 (over a 55-long period in 1965-2019) meteorological stations Germany and Netherlands by focusing on the exceptional drought in the year 2018.*

*The evaluation of this manuscript is based on the following questions:*

- *Is it a novel work based on a reliable scientific technique?*
- *Is it clearly structured and well-written?*
- *Are the experimental design and analysis of data adequate and appropriate to the investigation?*

*The manuscript is well-written and potentially interesting for HESS. It presents novel work on the extreme drought recorded in 2018 in continental Europe. Nonetheless, this manuscript requires substantial improvement before publication. The manuscript is not well-presented, model set up is oversimplified and data analysis is fair.*

Response: We very much appreciate the comments by the reviewer, which helped to improve the manuscript. We will revise the manuscript taking into account the comments.

*The main scientific question to the authors: is the 2018 drought an episodic event (as 1976 and 2003) or a consequence of a significant drying trend? From the abstract, the authors state that meteorological drought is episodic and SPI and other rainfall indeces indicate no significant declining trend while temperature-based ET is characterized by an increasing trend. The authors should quantify the probability related to this extreme episodic event from the SPI distribution. Same for the SSI as a consequence.*

*Lines 384-385: "In summary, the increase of droughts is mainly related to increasing soil moisture deficits, and reduction in actual ET. The main driver is not a precipitation decrease, but an increase of potential ET."*

Response: Thanks for pointing out this concern. We will add an analysis which will indicate how much more likely a 2018 drought is now, compared to the past, in terms of SPI, SSI, ET deficit and PPD.

*Abstract*

*Abstract is generally OK. In order to get any feedback or relationship between climate and hydrological response through the use of indeces referred to six months or to the growing season makes sense. I recommend to remove indeces referred to 12 month duration since it ignores the fundamental impact of rainfall seasonality.*

Response: Thanks for the suggestion by the reviewer. We prefer to keep indices for the 12-month duration period, as we see that especially 6-month duration drought in summer increases related to increased potential ET, whereas 12-month duration drought is less subject to change. With respect to the SPI and SSI trends in the figure 5, they are trends for the 12-month time period instead of the 6-month period. We will add 6-month SPI and SSI trends in the revised supplements and new text in the MS in an effort to clarify the use of the two time scales.

*Introduction*
*The state-of-the-art is well written. I list other interesting references on meteorological and agricultural droughts. Please see and comment about soil moisture index (Hunt et al., 2009)*

Response: Thanks. We will include the recommended references.

1. *Data and Methodology*

*The authors should specify that the grass-reference potential evapotranspiration, $ET_0$ is converted into crop-specific potential evapotranspiration, $ET_c$ by using a time-variant crop coefficient, $K_c$. Then, $ET_p$ is partitioned into potential transpiration, $T_p$ and potential evaporation, $E_p$ by using a time-variant leaf area index, LAI.*

Response: We will clarify this in the revised version of the manuscript and also include the equation used to calculate potential ET according Penman-Monteith, and the parameters adopted there.

*Moreover, root depth is time-variant is the crop is annual and root distribution across the root zone needs to be specified. Actual evaporation, $E_a$ and transpiration, $T_a$ are calculated in Hydrus-1D depending on soil surface and root zone pressure head values, respectively. It is therefore clear that the simulation of $ET_a$ depends on time-variant crop characteristics and local soil hydraulic properties. In lines 174-177 I understand that Kc is ignored, LAI and root depth are considered as 2.0 (or 2.88) and 50 cm, respectively. The soil hydraulic properties should be ideally measured. If direct measurements are not available, it is highly recommended to use PTFs based on silt, clay, sand contents, bulk density and organic matter (Weihermüller et al., 2021; Nasta et al., 2021). This study is basically a sensitivity analysis by considering that the spatial variability is quantified only in terms of soil texture classes (the van Genuchten's soil hydraulic parameters are crudely derived from tabulated values in Carsel and Parrish, 1988 in Table 3) and vegetation (assumed pastureland over the 31 stations) characteristics are constant in time and uniform in space.*

Response: The five soil types are representative for the domain because they cover well the soil texture triangle and for each location calculations are repeated for these five soil types, covering different possible conditions near the measurement sites. It can be expected that in a region

around a measurement site all these different soil types are present. In this study, our main objective is not to determine for each location as good as possible what the soil moisture and evapotranspiration conditions were in 2018 (and other years). This would require the use of all possible information sources including remote sensing information on soil moisture and vegetation states, precise soil and land use land cover information, among others. Our objective was to make a standardized comparison with past years and past droughts. For the further past, especially before the year 2000, remote sensing information is not of good quality, and a data based comparison is not possible. This is the reason why we used a model based comparison, covering all possible soil types.

The pasture was chosen because in this case it is known that all 31 meteorological stations are default located on a pasture. However, we agree with the reviewer that in a similar manner as for soil types, it would have been possible to cover different vegetation types (e.g., grassland, crop land, forest). To some limited extent we took different vegetation states into account by performing calculations with two different LAI time series. We decided not to include simulations for different vegetation types given the large amount of information already included in the paper, and also because we found that for the five different soil types, the drought trends and ranking of the drought years were hardly affected by soil type, in spite of the fact that absolute soil moisture contents showed large differences between soil types. This is also the reason why we did not consider time dependent root zone distributions. For the two LAI time series smaller differences among sites were observed, and the ranking was not affected. We already included a discussion on the impact of vegetation type on the results, pointing to the fact that for deep rooting vegetation rankings could be affected. However, we consider that our model-based comparison already covers many different conditions and think that a further extension is beyond the scope of this manuscript, and will not affect the main conclusions of this paper.

We will improve the clarification of the objectives in this paper, and specify that we want to perform a standardized comparison between years based on a model, to cover different conditions and in particular different soil types. However, a further extension to many different vegetation characteristics is beyond the scope of the manuscript and it is not expected that the main conclusions will be affected by it.

*In Eq. 2 remove P and ET from the Richards equation. P and $E_p$ are the climate forcings on the upper boundary. $T_p$ is reduced to Ta through the sink term, S in Eq. 2*

Response: Thanks, we will modify this.

1. *Results*

*Line 261: The strongest decrease in precipitation is in southern Germany (-2.2 mm/year) sounds really insignificant if compared to its mean annual value. From Fig. 2d, I see that the trend is from 800 mm/year (or so) to almost 700 mm/year (or so). Please explain.*

*Same problem for ET trends (Fig. 3)*

Response: We will clarify this in the manuscript and indicate that although a change of 2 mm/year seems to be small, this mounts to quite a large change over the considered time period.

*To tell the truth, I don't understand Fig. 7 and description of Fig. 7. Please, improve the presentation*

Thanks for the comment. We wanted to introduce this section with a general overview on how exceptional the year 2018 was from a meteorological point of view. We made the analysis on the basis of the ranking of the year 2018 in the complete time series of 55 years, for the different meteorological variables which influence drought. We will improve the description and discussion of Figure 7 in the text.

1. *Discussion*

*Please, be more critical and evidence if there is room for future improvements*

Response: We will extend the discussion and will discuss the limitations of the model, and the possibilities to improve the analysis in the future with different model types. We will also discuss additional variables which could be analyzed with more complex models, like for example crop yield. On the other hand, we will discuss in the revised version the limitations of such a more complex, integrated model.

*References*

*Hein, A., Condon, L., Maxwell, R. 2019. Evaluating the relative importance of precipitation, temperature and land-cover change in the hydrologic response to extreme meteorological drought conditions over the North American High PlainsHydrol. Earth Syst. Sci., 23, 1931–1950, 2019*

*Hunt, E.D., K.G. Hubbard, D.A. Wilhite, T.J. Arkebauer, and A.L. Dutcher. 2009. The development and evaluation of a soil moisture index. Int. J. Climatol. 29:747–759. doi:10.1002/joc.1749*

*Martínez-Fernández, J., González-Zamora, A., Gamuzzio, A. 2015. A soil water based index as a suitable agricultural drought indicator. Journal of Hydrology 522, 265–273*

*Nasta P., B. Szabó, N. Romano. 2021. Evaluation of Pedotransfer Functions for predicting soil hydraulic properties: A voyage from regional to field scales across Europe. Journal of Hydrology: Regional Studies 37, https://doi.org/10.1016/j.ejrh.2021.100903*

Sánchez, N., Á. González-Zamora, M. Piles and J. Martínez-Fernández. 2016. A New Soil Moisture Agricultural Drought Index (SMADI) Integrating MODIS and SMOS Products: A Case of Study over the Iberian Peninsula. Remote Sensing. 8, 287; doi:10.3390/rs8040287

Van Loon, A.F. 2015. Hydrological drought explained. WIREs Water 2015, 2:359–392. doi: 10.1002/wat2.1085

von Gunten, D., T. Wöhling, C. P. Haslauer, D. Merchán, J. Causapé, and O. A. Cirpka. 2016. Using an integrated hydrological model to estimate the usefulness of meteorological drought indices in a changing climate. Hydrol. Earth Syst. Sci., 20, 4159–4175

Weihermüller, L., Lehmann, P., Herbst, M., Rahmati, M., Verhoef, A., Or, D., et al. (2021). Choice of pedotransfer functions matters when simulating soil water balance fluxes. Journal of Advances in Modeling Earth Systems, 13, e2020MS002404. https://doi.org/10.1029/2020MS002404

---

## Author Comment (AC4)

**Response (in blue) to comments by Anonymous Referee #3**

The main idea of this study is to assess how extreme the drought was in the year 2018 in Netherlands and Germany. They used Hydrus-1D model to simulate soil moisture, actual ET. Based on this and together with the meteorological station data, they calculated SPI, SSI, PPD and ET deficit to analyze the drought.

This topic is interesting and the manuscript is well-written. However, this manuscript needs some revision before publication. There are some questions to the authors:

Response: We very much thank the reviewer for the comments, which helped to improve the manuscript. We will revise the manuscript taking into account the comments.

1) Data and Methodology

Q1: Do you have insitu measurements to do validation about the HYDRUS 1-D model simulation? If not, how could you be sure the simulation results (actual ET, soil moisture) meet the evaluation requirements in statistics, for example, RMSE, pearson correlation coefficient. In lines 90-96, you mentioned in situ measured soil moisture data and remotely sensed soil moisture are not available for such long time series and are in general strongly affected by measurement uncertainties. But at least you should have some in situ measured soil moisture and actual ET to do the validation of the HYDRUS 1-D model simulation.

Response: Thanks for the suggestion. This would have been good, but as explained in our response to reviewer #1 our aim is a standardized model-based comparison where we cover many different soil types. Covering very different soil types guarantees us that we account for the different possible soil types at and around the measurement sites. We will clarify this further in the paper, and also stress that our objective is a model-based comparison between droughts in different years which covers different possible conditions. The objective is not to reproduce as precisely as possible the specific soil moisture and evapotranspiration at a site for a given year. With the (very) different soil types we cover the different soils which are potentially available around the sites.

Q2: In lines 174-177, please give the equation and explain how did you derive the parameters of Penman-Monteith equation.

Response: Thanks for the suggestion. We will add new text and include the equation to calculate potential ET and the different parameter settings used.

Q3: Why the pasture is assumed for all these stations? For these 31 stations, do you have the vegetation types information? You should use them in the simulation.

Response: Thanks for pointing this out. All meteorological stations are located on pasture, which is standard for meteorological stations operated according WMO (World Meteorological Organization) regulation. We will clarify this in the paper.

Q4: In lines 85-87, you used five different soil types out of 12 textural soil classes. How did you determine these five soil types? Do you have the soil types information of these 31 stations?

Response: Thanks for pointing this out. The studied five soils cover well the soil textural triangle, so that we cover very different soil hydraulic parameters. Unfortunately, we do not have soil information for the climatic stations. However, covering very different soil types guarantees us that we account for the different possible soil types at and around the measurement sites. We will clarify this further in the paper, and also stress that our objective is a model-based comparison between years which covers different possible conditions. The objective is not to reproduce as precisely as possible the specific soil moisture and evapotranspiration at a site for a given year.

2)Results

Q1: The trend analysis of each variable should be more in-depth. The summary of part 3.1 is one sentence. The in-depth analysis and summary should be done based on the trends of multiple variables, combined with physical processes. For different sites, you should analyze the potential causes that may cause severe drought based on the specific local geographic environment. Probably you can put it in the discussion part, but at least you need to analyze it.

Response: Thanks for this constructive suggestion, which will help to improve the manuscript. We will extend the discussion in the revised manuscript covering this.

Q2: For the trend figure, I did not see the significance test result although you mentioned MK can do it in lines 250-252.

Response: Thanks for this suggestion. We will include confidence intervals for the trend lines for all figures.

Q3: Logically, I did not understand the connection between the section 3.1 and section 3.2. You need to strengthen the logical connection.

Response: Thanks for the suggestion. We will add new text to the beginning of section 3.2 in the revised manuscript to provide a more logical connection between the two sections.

Q4: To be honest, I do not understand what do you want to say in figure7. The description needs to be improved.

Response: Thanks for the comment. We wanted to introduce this section with a general overview on how exceptional the year 2018 was from a meteorological point of view. We made the analysis on the basis of the ranking of the year 2018 in the complete time series of 55 years, for the different meteorological variables which influence drought. We will improve the description of Figure 7 in the text.

3)Discussion

You need to analyze more in this section.

Response: Thanks for the suggestion. We will add discussion.

Firstly, for the model set up, you need to point out the potential for further improvement.

Response: Thanks. We agree that we could discuss this in more detail. We will add additional discussion for the model setup and the limitations of it.

Secondly, for the analysis part, you used precipitation, potential ET, actual ET, soil moisture, and four drought indices for drought trend analysis. There are other variables could be considered as well from the physical process point of view, please describe more about the future improvements.

Response: Thanks for the suggestion. We think that the most important variables are already included here. An extension could be a coupled soil hydrological-crop model, with the assessment of crop yield. For the assessment of other hydrological variables discharge and groundwater levels also need to be simulated with an integrated hydrological or coupled land surface-subsurface model. We will add additional discussion in the manuscript.